# vGraph: A Generative Model for Joint Community Detection and Node Representation Learning

**Fan-Yun Sun**[1,2], **Meng Qu**[2], **Jordan Hoffmann**[2,3], **Chin-Wei Huang**[2,4], **Jian Tang**[2,5,6]
[1]National Taiwan University,
[2]Mila-Quebec Institute for Learning Algorithms, Canada
[3]Harvard University, USA
[4]Element AI, Canada
[5]HEC Montreal, Canada
[6]CIFAR AI Research Chair
b04902045@ntu.edu.tw

## Abstract

This paper focuses on two fundamental tasks of graph analysis: community detection and node representation learning, which capture the global and local structures of graphs, respectively. In the current literature, these two tasks are usually independently studied while they are actually highly correlated. We propose a probabilistic generative model called vGraph to learn community membership and node representation collaboratively. Specifically, we assume that each node can be represented as a mixture of communities, and each community is defined as a multinomial distribution over nodes. Both the mixing coefficients and the community distribution are parameterized by the low-dimensional representations of the nodes and communities. We designed an effective variational inference algorithm which regularizes the community membership of neighboring nodes to be similar in the latent space. Experimental results on multiple real-world graphs show that vGraph is very effective in both community detection and node representation learning, outperforming many competitive baselines in both tasks. We show that the framework of vGraph is quite flexible and can be easily extended to detect hierarchical communities.

## 1 Introduction

Graphs, or networks, are a general and flexible data structure to encode complex relationships among objects. Examples of real-world graphs include social networks, airline networks, protein-protein interaction networks, and traffic networks. Recently, there has been increasing interest from both academic and industrial communities in analyzing graphical data. Examples span a variety of domains and applications such as node classification [3, 26] and link prediction [8, 32] in social networks, role prediction in protein-protein interaction networks [16], and prediction of information diffusion in social and citation networks [22].

One fundamental task of graph analysis is community detection, which aims to cluster nodes into multiple groups called communities. Each community is a set of nodes that are more closely connected to each other than to nodes in different communities. A community level description is able to capture important information about a graph's global structure. Such a description is useful in many real-world applications, such as identifying users with similar interests in social networks [22] or proteins with similar functionality in biochemical networks [16]. Community detection has been extensively studied in the literature, and a number of methods have been proposed, including algorithmic approaches [1, 5] and probabilistic models [10, 20, 36, 37]. A classical approach to

detect communities is spectral clustering [34], which assumes that neighboring nodes tend to belong to the same communities and detects communities by finding the eigenvectors of the graph Laplacian.

Another important task of graph analysis is node representation learning, where nodes are described using low-dimensional features. Node representations effectively capture local graph structure and are often used as features for many prediction tasks. Modern methods for learning node embeddings [11, 24, 26] have proved effective on a variety of tasks such as node classification [3, 26], link prediction [8, 32] and graph visualization [27, 31].

Clustering, which captures the global structure of graphs, and learning node embeddings, which captures local structure, are typically studied separately. Clustering is often used for exploratory analysis, while generating node embeddings is often done for predictive analysis. However, these two tasks are very correlated and it may be beneficial to perform both tasks simultaneously. The intuition is that (1) node representations can be used as good features for community detection (e.g., through K-means) [4, 25, 29], and (2) the node community membership can provide good contexts for learning node representations [33]. However, how to leverage the relatedness of node clustering and node embedding in a unified framework for joint community detection and node representation learning is under-explored.

In this paper, we propose a novel probabilistic generative model called vGraph for joint community detection and node representation learning. vGraph assumes that each node $v$ can be represented as a mixture of multiple communities and is described by a multinomial distribution over communities $z$, i.e., $p(z|v)$. Meanwhile, each community $z$ is modeled as a distribution over the nodes $v$, i.e., $p(v|z)$. vGraph models the process of generating the neighbors for each node. Given a node $u$, we first draw a community assignment $z$ from $p(z|u)$. This indicates which community the node is going to interact with. Given the community assignment $z$, we generate an edge $(u, v)$ by drawing another node $v$ according to the community distribution $p(v|z)$. Both the distributions $p(z|v)$ and $p(v|z)$ are parameterized by the low-dimensional representations of the nodes and communities. As a result, this approach allows the node representations and the communities to interact in a mutually beneficial way. We also design a very effective algorithm for inference with backpropagation. We use variational inference for maximizing the lower-bound of the data likelihood. The Gumbel-Softmax [13] trick is leveraged since the community membership variables are discrete. Inspired by existing spectral clustering methods [6], we added a smoothness regularization term to the objective function of the variational inference routine to ensure that community membership of neighboring nodes is similar. The whole framework of vGraph is very flexible and general. We also show that it can be easily extended to detect hierarchical communities.

In the experiment section, we show results on three tasks: overlapping community detection, non-overlapping community detection, and node classification– all using various real-world datasets. Our results show that vGraph is very competitive with existing state-of-the-art approaches for these tasks. We also present results on hierarchical community detection. Relevant source codes have been made public [1].

## 2  Related Work

**Community Detection.** Many community detection methods are based on matrix factorization techniques. Typically, these methods try to recover the node-community affiliation matrix by performing a low-rank decomposition of the graph adjacency matrix or other related matrices [17, 18, 32, 36]. These methods are not scalable due to the complexity of matrix factorization, and their performance is restricted by the capacity of the bi-linear models. Many other studies develop generative models for community detection. Their basic idea is to characterize the generation process of graphs and cast community detection as an inference problem [37, 38, 39]. However, the computational complexity of these methods is also high due to complicated inference. Compared with these approaches, vGraph is more scalable and can be efficiently optimized with backpropagation and Gumbel-Softmax [13, 19]. Additionally, vGraph is able to learn and leverage the node representations for community detection.

**Node Representation Learning.** The goal of node representation learning is to learn distributed representations of nodes in graphs so that nodes with similar local connectivity tend to have sim-

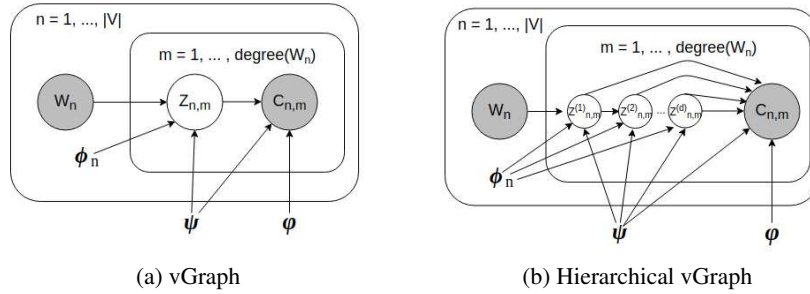

(a) vGraph             (b) Hierarchical vGraph

Figure 1: The diagram on the left represents the graphical model of vGraph and the diagram on the right represents the graphical model of the hierarchical extension. $\phi_n$ is the embedding of node $w_n$, $\psi$ denotes the embedding of communities, and $\varphi$ denotes the embeddings of nodes used in $p(c|z)$. Refer to Eq. 2 and Eq. 3.

ilar representations. Some representative methods include DeepWalk [24], LINE [26], node2vec [11] and GraphRep [3]. Typically, these methods explore the local connectivity of each node by conducting random walks with either breadth-first search [24] or depth-first search [26]. Despite their effectiveness in a variety of applications, these methods mainly focus on preserving the local structure of graphs, therefore ignoring global community information. In vGraph, we address this limitation by treating the community label as a latent variable. This way, the community label can provide additional contextual information which enables the learned node representations to capture the global community information.

**Framework for node representation learning and community detection.** There exists previous work [4, 14, 29, 30, 33] that attempts to solve community detection and node representation learning jointly. However, their optimization process alternates between community assignment and node representation learning instead of simultaneously solving both tasks [4, 30]. Compared with these methods, vGraph is scalable and the optimization is done end-to-end.

**Mixture Models.** Methodologically, our method is related to mixture models, particularly topic models (e.g. PSLA [12] and LDA [2]). These methods simulate the generation of words in documents, in which topics are treated as latent variables, whereas we consider generating neighbors for each node in a graph, and the community acts as a latent variable. Compared with these methods, vGraph parameterizes the distributions with node and community embeddings, and all the parameters are trained with backpropagation.

## 3 Problem Definition

Graphs are ubiquitous in the real-world. Two fundamental tasks on graphs are community detection and learning node embeddings, which focus on global and local graph structures respectively and hence are naturally complementary. In this paper, we study jointly solving these two tasks. Let $\mathcal{G} = (\mathcal{V}, \mathcal{E})$ represent a graph, where $\mathcal{V} = \{v_1, \ldots, v_V\}$ is a set of vertices and $\mathcal{E} = \{e_{ij}\}$ is the set of edges. Traditional graph embedding aims to learn a node embedding $\phi_i \in \mathbb{R}^d$ for each $v_i \in \mathcal{V}$ where $d$ is predetermined. Community detection aims to extract the community membership $\mathcal{F}$ for each node. Suppose there are $K$ communities on the graph $\mathcal{G}$, we can denote the community assignment of node $v_i$ as $\mathcal{F}(v_i) \subseteq \{1, ..., K\}$. We aim to jointly learn node embeddings $\phi$ and community affiliation of vertices $\mathcal{F}$.

## 4 Methodology

In this section, we introduce our generative approach vGraph, which aims at collaboratively learning node representations and detecting node communities. Our approach assumes that each node can belong to multiple communities representing different social contexts [7]. Each node should generate different neighbors under different social contexts. vGraph parameterizes the node-community distributions by introducing node and community embeddings. In this way, the node representations

can benefit from the detection of node communities. Similarly, the detected community assignment can in turn improve the node representations. Inspired by existing spectral clustering methods [6], we add a smoothness regularization term that encourages linked nodes to be in the same communities.

## 4.1  vGraph

vGraph models the generation of node neighbors. It assumes that each node can belong to multiple communities. For each node, different neighbors will be generated depending on the community context. Based on the above intuition, we introduce a prior distribution $p(z|w)$ for each node $w$ and a node distribution $p(c|z)$ for each community $z$. The generative process of each edge $(w, c)$ can be naturally characterized as follows: for node $w$, we first draw a community assignment $z \sim p(z|w)$, representing the social context of $w$ during the generation process. Then, the linked neighbor $c$ is generated based on the assignment $z$ through $c \sim p(c|z)$. Formally, this generation process can be formulated in a probabilistic way:

$$p(c|w) = \sum_z p(c|z)p(z|w). \tag{1}$$

vGraph parameterizes the distributions $p(z|w)$ and $p(c|z)$ by introducing a set of node embeddings and community embeddings. Note that different sets of node embeddings are used to parametrize the two distributions. Specifically, let $\phi_i$ denote the embedding of node $i$ used in the distribution $p(z|w)$, $\varphi_i$ denote the embedding of node $i$ used in $p(c|z)$, and $\psi_j$ denote the embedding of the $j$-th community. The prior distribution $p_{\phi,\psi}(z|w)$ and the node distribution conditioned on a community $p_{\psi,\varphi}(c|z)$ are parameterized by two softmax models:

$$p_{\phi,\psi}(z=j|w) = \frac{\exp(\phi_w^T \psi_j)}{\sum_{i=1}^{K} \exp(\phi_w^T \psi_i)}, \tag{2}$$

$$p_{\psi,\varphi}(c|z=j) = \frac{\exp(\psi_j^T \varphi_c)}{\sum_{c' \in \mathcal{V}} \exp(\psi_j^T \varphi_{c'})}. \tag{3}$$

Calculating Eq. 3 can be expensive as it requires summation over all vertices. Thus, for large datasets we can employ negative sampling as done in LINE [26] using the following objective function:

$$\log \sigma(\varphi_c^T \cdot \psi_j) + \sum_{i=1}^{K} E_{v \sim P_n(v)}[\log \sigma(-\varphi_v^T \cdot \psi_j)], \tag{4}$$

where $\sigma(x) = 1/(1 + \exp(-x))$, $P_n(v)$ is a noise distribution, and $K$ is the number of negative samples. This, combined with stochastic optimization, enables our model to be scalable.

To learn the parameters of vGraph, we try to maximize the log-likelihood of the observed edges, i.e., $\log p_{\phi,\varphi,\psi}(c|w)$. Since directly optimizing this objective is intractable for large graphs, we instead optimize the following evidence lower bound (ELBO) [15]:

$$\mathcal{L} = E_{z \sim q(z|c,w)}[\log p_{\psi,\varphi}(c|z)] - \mathrm{KL}(q(z|c,w)||p_{\phi,\psi}(z|w)) \tag{5}$$

where $q(z|c, w)$ is a variational distribution that approximates the true posterior distribution $p(z|c, w)$, and $\mathrm{KL}(\cdot||\cdot)$ represents the Kullback-Leibler divergence between two distributions.

Specifically, we parametrize the variational distribution $q(z|c, w)$ with a neural network as follows:

$$q_{\phi,\psi}(z=j|w,c) = \frac{\exp((\phi_w \odot \phi_c)^T \psi_j)}{\sum_{i=1}^{K} \exp((\phi_w \odot \phi_c)^T \psi_i)}. \tag{6}$$

where $\odot$ denotes element-wise multiplication. We chose element-wise multiplication because it is symmetric and it forces the representation of the edge to be dependent on both nodes.

The variational distribution $q(z|c, w)$ represents the community membership of the edge $(w, c)$. Based on this, we can easily approximate the community membership distribution of each node $w$, i.e., $p(z|w)$ by aggregating all its neighbors:

$$p(z|w) = \sum_c p(z, c|w) = \sum_c p(z|w, c)p(c|w) \approx \frac{1}{|N(w)|} \sum_{c \in N(w)} q(z|w, c), \tag{7}$$

where $N(w)$ is the set of neighbors of node $w$. To infer non-overlapping communities, we can simply take the $\arg\max$ of $p(z|w)$. However, when detecting overlapping communities instead of thresholding $p(z|w)$ as in [14], we use

$$\mathcal{F}(w) = \{\arg\max_k q(z = k|w, c)\}_{c \in N(w)}. \tag{8}$$

That is, we assign each edge to one community and then map the edge communities to node communities by gathering nodes incident to all edges within each edge community as in [1].

**Complexity.** Here we show the complexity of vGraph. Sampling an edge takes constant time, thus calculating Eq. (4) takes $\mathcal{O}(d(M+1))$ time, where $M$ is the number of negative samples and $d$ is the dimension of embeddings (the node embeddings and community embeddings have the same dimension). To calculate Eq. (6), it takes $\mathcal{O}(dK)$ time where $K$ is the number of communities. Thus, an iteration with one sample takes $\mathcal{O}(\max(dM, dK))$ time. In practice the number of updates required is proportional to the number of edges $\mathcal{O}(|\mathcal{E}|)$, thus the overall time complexity of vGraph is $\mathcal{O}(|\mathcal{E}|d\max(M, K))$.

## 4.2 Community-smoothness Regularized Optimization

For optimization, we need to optimize the lower bound (5) w.r.t. the parameters in the variational distribution and the generative parameters. If $z$ is continuous, the reparameterization trick [15] can be used. However, $z$ is discrete in our case. In principle, we can still estimate the gradient using a score function estimator [9, 35]. However, the score function estimator suffers from a high variance, even when used with a control variate. Thus, we use the Gumbel-Softmax reparametrization [13, 19] to obtain gradients for the evidence lower bound. More specifically, we use the straight-through Gumbel-Softmax estimator [13].

A community can be defined as a group of nodes that are more similar to each other than to those outside the group [23]. For a non-attributed graph, two nodes are similar if they are connected and share similar neighbors. However, vGraph does not explicitly weight local connectivity in this way. To resolve this, inspired by existing spectral clustering studies [6], we augment our training objective with a smoothness regularization term that encourages the learned community distributions of linked nodes to be similar. Formally, the regularization term is given below:

$$\mathcal{L}_{reg} = \lambda \sum_{(w,c)\in\mathcal{E}} \alpha_{w,c} \cdot d(p(z|c), p(z|w)) \tag{9}$$

where $\lambda$ is a tunable hyperparameter , $\alpha_{w,c}$ is a regularization weight, and $d(\cdot, \cdot)$ is the distance between two distributions (squared difference in our experiments). Motivated by [25], we set $\alpha_{w,c}$ to be the Jaccard's coefficient of node $w$ and $c$, which is given by:

$$\alpha_{w,c} = \frac{|N(w) \cap N(c)|}{|N(w) \cup N(c)|}, \tag{10}$$

where $N(w)$ denotes the set of neighbors of $w$. The intuition behind this is that $\alpha_{w,c}$ serves as a similarity measure of how similar the neighbors are between two nodes. Jaccard's coefficient is used for this metric and thus the higher the value of Jaccard's coefficient, the more the two nodes are encouraged to have similar distribution over communities.

By combining the evidence lower bound and the smoothness regularization term, the entire loss function we aim to minimize is given below:

$$\mathcal{L} = -E_{z \sim q_{\phi,\psi}(z|c,w)}[\log p_{\psi,\varphi}(c|z)] + \text{KL}(q_{\phi,\psi}(z|c,w)||p_{\phi,\psi}(z|w)) + \mathcal{L}_{reg} \tag{11}$$

For large datasets, negative sampling can be used for the first term.

## 4.3 Hierarchical vGraph

One advantage of vGraph's framework is that it is very general and can be naturally extended to detect hierarchical communities. In this case, suppose we are given a $d$-level tree and each node is associate with a community, the community assignment can be represented as a $d$-dimensional path vector $\vec{z} = (z^{(1)}, z^{(2)}, ..., z^{(d)})$, as shown in Fig. 1. Then, the generation process is formulated as

below: (1) a tree path $\vec{z}$ is sampled from a prior distribution $p_{\phi,\psi}(\vec{z}|w)$. (2) The context $c$ is decoded from $\vec{z}$ with $p_{\psi,\varphi}(c|\vec{z})$. Under this model, the likelihood of the network is

$$p_{\phi,\varphi,\psi}(c|w) = \sum_{\vec{z}} p_{\phi,\psi}(c|\vec{z})p_{\phi,\psi}(\vec{z}|w). \tag{12}$$

At every node of the tree, there is an embedding vector associated with the community. Such a method is similar to the hierarchical softmax parameterization used in language models [21].

## 5 Experiments

As vGraph can detect both overlapping and non-overlapping communities, we evaluate it on three tasks: overlapping community detection, non-overlapping community detection, and vertex classification.

### 5.1 Datasets

We evaluate vGraph on 20 standard graph datasets. For non-overlapping community detection and node classification, we use 6 datasets: Citeseer, Cora, Cornell, Texas, Washington, and Wisconsin. For overlapping communtiy detection, we use 14 datasets, including Facebook, Youtube, Amazon, Dblp, Coauthor-CS. For Youtube, Amazon, and Dblp, we consider subgraphs with the 5 largest ground-truth communities due to the runtime of baseline methods. To demonstrate the scalability of our method, we additionally include visualization results on a large dataset – Dblp-full. Dataset statistics are provided in Table 1. More details about the datasets is provided in Appendix A.

Table 1: Dataset Statistics. $|\mathcal{V}|$: number of nodes, $|\mathcal{E}|$: number of edges, $K$: number of communities, AS: average size of communities, AN: average number of communities that a node belongs to.

| Dataset | $|\mathcal{V}|$ | $|\mathcal{E}|$ | $K$ | AS | AN |
|---|---|---|---|---|---|
| Nonoverlapping | | | | | |
| Cornell | 195 | 286 | 5 | 39.00 | 1 |
| Texas | 187 | 298 | 5 | 37.40 | 1 |
| Washington | 230 | 417 | 5 | 46.00 | 1 |
| Wisconsin | 265 | 479 | 5 | 53.00 | 1 |
| Cora | 2708 | 5278 | 7 | 386.86 | 1 |
| Citeseer | 3312 | 4660 | 6 | 552.00 | 1 |
| overlapping | | | | | |
| facebook0 | 333 | 2519 | 24 | 13.54 | 0.98 |
| facebook107 | 1034 | 26749 | 9 | 55.67 | 0.48 |
| facebook1684 | 786 | 14024 | 17 | 45.71 | 0.99 |
| facebook1912 | 747 | 30025 | 46 | 23.15 | 1.43 |
| facebook3437 | 534 | 4813 | 32 | 6.00 | 0.36 |
| facebook348 | 224 | 3192 | 14 | 40.50 | 2.53 |
| facebook3980 | 52 | 146 | 17 | 3.41 | 1.12 |
| facebook414 | 150 | 1693 | 7 | 25.43 | 1.19 |
| facebook686 | 168 | 1656 | 14 | 34.64 | 2.89 |
| facebook698 | 61 | 270 | 13 | 6.54 | 1.39 |
| Youtube | 5346 | 24121 | 5 | 1347.80 | 1.26 |
| Amazon | 794 | 2109 | 5 | 277.20 | 1.75 |
| Dblp | 24493 | 89063 | 5 | 5161.40 | 1.05 |
| Coauthor-CS | 9252 | 33261 | 5 | 2920.60 | 1.58 |
| Dblp-full | 93432 | 335520 | 5000 | 22.45 | 1.20 |

### 5.2 Evaluation Metric

For overlapping community detection, we use *F1-Score* and *Jaccard Similarity* to measure the performance of the detected communities as in [37, 18]. For non-overlapping community detection, we use *Normalized Mutual Information (NMI)* [28] and *Modularity*. Note that Modularity does not utilize ground truth data. For node classification, *Micro-F1* and *Macro-F1* are used.

### 5.3 Comparative Methods

For overlapping community detection, we choose four competitive baselines: **BigCLAM** [36], a nonnegative matrix factorization approach based on the Bernoulli-Poisson link that only considers the graph structure; **CESNA** [37], an extension of BigCLAM, that additionally models the generative process for node attributes; **Circles** [20], a generative model of edges w.r.t. attribute similarity to detect communities; and **SVI** [10], a Bayesian model for graphs with overlapping communities that uses a mixed-membership stochastic blockmodel.

To evaluate node embedding and non-overlapping community detection, we compare our method with the five baselines: **MF** [32], which represents each vertex with a low-dimensional vector obtained through factoring the adjacency matrix; **DeepWalk** [24], a method that adopts truncated random walk and Skip-Gram to learn vertex embeddings; **LINE** [26], which aims to preserve the first-order and second-order proximity among vertices in the graph; **Node2vec** [11], which adopts biased random walk and Skip-Gram to learn vertex embeddings; and **ComE** [4], which uses a Gaussian mixture model to learn an embedding and clustering jointly using random walk features.

### 5.4 Experiment Configuration

For all baseline methods, we use the implementations provided by their authors and use the default parameters. For methods that only output representations of vertices, we apply K-means to the

Table 2: Evaluation (in terms of F1-Score and Jaccard Similarity) on networks with overlapping ground-truth communities. NA means the task is not completed in 24 hours. In order to evaluate the effectiveness of smoothness regularization, we show the result of our model with (vGraph+) and without the regularization.

| | F1-score | | | | | | Jaccard | | | | | |
|---|---|---|---|---|---|---|---|---|---|---|---|---|
| Dataset | Bigclam | CESNA | Circles | SVI | vGraph | vGraph+ | Bigclam | CESNA | Circles | SVI | vGraph | vGraph+ |
| facebook0 | **0.2948** | 0.2806 | 0.2860 | 0.2810 | 0.2440 | 0.2606 | **0.1846** | 0.1725 | 0.1862 | 0.1760 | 0.1458 | 0.1594 |
| facebook107 | **0.3928** | 0.3733 | 0.2467 | 0.2689 | 0.2817 | 0.3178 | **0.2752** | 0.2695 | 0.1547 | 0.1719 | 0.1827 | 0.2170 |
| facebook1684 | 0.5041 | **0.5121** | 0.2894 | 0.3591 | 0.4232 | 0.4379 | 0.3801 | **0.3871** | 0.1871 | 0.2467 | 0.2917 | 0.3272 |
| facebook1912 | 0.3493 | 0.3474 | 0.2617 | 0.2804 | 0.2579 | **0.3750** | 0.2412 | 0.2394 | 0.1672 | 0.2010 | 0.1855 | **0.2796** |
| facebook3437 | 0.1986 | 0.2009 | 0.1009 | 0.1544 | 0.2087 | **0.2267** | 0.1148 | 0.1165 | 0.0545 | 0.0902 | 0.1201 | **0.1328** |
| facebook348 | 0.4964 | 0.5375 | 0.5175 | 0.4607 | **0.5539** | 0.5314 | 0.3586 | 0.4001 | 0.3927 | 0.3360 | **0.4099** | 0.4050 |
| facebook3980 | 0.3274 | 0.3574 | 0.3203 | NA | **0.4450** | 0.4150 | 0.2426 | 0.2645 | 0.2097 | NA | **0.3376** | 0.2933 |
| facebook414 | 0.5886 | 0.6007 | 0.4843 | 0.3893 | 0.6471 | **0.6693** | 0.4713 | 0.4732 | 0.3418 | 0.2931 | 0.5184 | **0.5587** |
| facebook686 | 0.3825 | 0.3900 | 0.5036 | 0.4639 | 0.4775 | **0.5379** | 0.2504 | 0.2534 | 0.3615 | 0.3394 | 0.3272 | **0.3856** |
| facebook698 | 0.5423 | 0.5865 | 0.3515 | 0.4031 | 0.5396 | **0.5950** | 0.4192 | 0.4588 | 0.2255 | 0.3002 | 0.4356 | **0.4771** |
| Youtube | 0.4370 | 0.3840 | 0.3600 | 0.4140 | 0.5070 | **0.5220** | 0.2929 | 0.2416 | 0.2207 | 0.2867 | 0.3434 | **0.3480** |
| Amazon | 0.4640 | 0.4680 | **0.5330** | 0.4730 | **0.5330** | 0.5320 | 0.3505 | 0.3502 | 0.3671 | 0.3643 | 0.3689 | **0.3693** |
| Dblp | 0.2360 | 0.3590 | NA | NA | 0.3930 | **0.3990** | 0.1384 | 0.2226 | NA | NA | 0.2501 | **0.2505** |
| Coauthor-CS | 0.3830 | 0.4200 | NA | 0.4070 | 0.4980 | **0.5020** | 0.2409 | 0.2682 | NA | 0.2972 | **0.3517** | 0.3432 |

learned embeddings to get non-overlapping communities. Results report are averaged over 5 runs. **No node attributes** are used in all our experiments. We generate node attributes using node degree features for those methods that require node attributes such as CESNA [37] and Circles [20]. It is hard to compare the quality of community results when the numbers of communities are different for different methods. Therefore, we set the number of communities to be detected, $K$, as the number of ground-truth communities for all methods, as in [18]. For vGraph, we use full-batch training when the dataset is small enough. Otherwise, we use stochastic training with a batch size of 5000 or 10000 edges. The initial learning rate is set to 0.05 and is decayed by 0.99 after every 100 iterations. We use the Adam optimizer and we trained for 5000 iterations. When smoothness regularization is used, $\lambda$ is set to 100. For community detection, the model with the lowest loss is chosen. For node classification, we evaluate node embeddings after 1000 iterations of training. The dimension of node embeddings is set to 128 in all experiments for all methods. For the node classification task, we randomly select 70% of the labels for training and use the rest for testing.

## 5.5   Results

Table 2 shows the results on overlapping community detection. Some of the methods are not very scalable and cannot obtain results in 24 hours on some larger datasets. Compared with these studies, vGraph outperforms all baseline methods in 11 out of 14 datasets in terms of F1-score or Jaccard Similarity, as it is able to leverage useful representations at node level. Moreover, vGraph is also very efficient on these datasets, since we use employ variational inference and parameterize the model with node and community embeddings. By adding the smoothness regularization term (vGraph+), we see a farther increase performance, which shows that our method can be combined with concepts from traditional community detection methods.

The results for non-overlapping community detection are presented in Table 3. vGraph outperforms all conventional node embeddings + K-Means in 4 out of 6 datasets in terms of NMI and outperforms all 6 in terms of modularity. ComE, another framework that jointly solves node embedding and community detection, also generally performs better than other node embedding methods + K-Means. This supports our claim that learning these two tasks collaboratively instead of sequentially can further enhance performance. Compare to ComE, vGraph performs better in 4 out of 6 datasets in terms of NMI and 5 out of 6 datasets in terms of modularity. This shows that vGraph can also outperform frameworks that learn node representations and communities together.

Table 4 shows the result for the node classification task. vGraph significantly outperforms all the baseline methods in 9 out of 12 datasets. The reason is that most baseline methods only consider the local graph information without modeling the global semantics. vGraph solves this problem by representing node embeddings as a mixture of communities to incorporate global context.

Table 3: Evaluation (in terms of NMI and Modularity) on networks with non-overlapping ground-truth communities.

| | NMI | | | | | | Modularity | | | | | |
|---|---|---|---|---|---|---|---|---|---|---|---|---|
| Dataset | MF | deepwalk | LINE | node2vec | ComE | vGraph | MF | deepwalk | LINE | node2vec | ComE | vGraph |
| cornell | 0.0632 | 0.0789 | 0.0697 | 0.0712 | 0.0732 | **0.0803** | 0.4220 | 0.4055 | 0.2372 | 0.4573 | 0.5748 | **0.5792** |
| texas | 0.0562 | 0.0684 | **0.1289** | 0.0655 | 0.0772 | 0.0809 | 0.2835 | 0.3443 | 0.1921 | 0.3926 | **0.4856** | 0.4636 |
| washington | 0.0599 | 0.0752 | **0.0910** | 0.0538 | 0.0504 | 0.0649 | 0.3679 | 0.1841 | 0.1655 | 0.4311 | 0.4862 | **0.5169** |
| wisconsin | 0.0530 | 0.0759 | 0.0680 | 0.0749 | 0.0689 | **0.0852** | 0.3892 | 0.3384 | 0.1651 | 0.5338 | 0.5500 | **0.5706** |
| cora | 0.2673 | 0.3387 | 0.2202 | 0.3157 | **0.3660** | 0.3445 | 0.6711 | 0.6398 | 0.4832 | 0.5392 | 0.7010 | **0.7358** |
| citeseer | 0.0552 | 0.1190 | 0.0340 | 0.1592 | **0.2499** | 0.1030 | 0.6963 | 0.6819 | 0.4014 | 0.4657 | 0.7324 | **0.7711** |

Table 4: Results of node classification on 6 datasets.

| | Macro-F1 | | | | | | Micro-F1 | | | | | |
|---|---|---|---|---|---|---|---|---|---|---|---|---|
| Datasets | MF | DeepWalk | LINE | Node2Vec | ComE | vGraph | MF | DeepWalk | LINE | Node2Vec | ComE | vGraph |
| Cornell | 13.05 | 22.69 | 21.78 | 20.70 | 19.86 | **29.76** | 15.25 | 33.05 | 23.73 | 24.58 | 25.42 | **37.29** |
| Texas | 8.74 | 21.32 | 16.33 | 14.95 | 15.46 | **26.00** | 14.03 | 40.35 | 27.19 | 25.44 | 33.33 | **47.37** |
| Washington | 15.88 | 18.45 | 13.99 | 21.23 | 15.80 | **30.36** | 15.94 | 34.06 | 25.36 | 28.99 | 33.33 | **34.78** |
| Wisconsin | 14.77 | 23.44 | 19.06 | 18.47 | 14.63 | **29.91** | 18.75 | **38.75** | 28.12 | 25.00 | 32.50 | 35.00 |
| Cora | 11.29 | 13.21 | 11.86 | 10.52 | 12.88 | **16.23** | 12.79 | 22.32 | 14.59 | 27.74 | **28.04** | 24.35 |
| Citeseer | 14.59 | 16.17 | 15.99 | 16.68 | 12.88 | **17.88** | 15.79 | 19.01 | 16.80 | **20.82** | 19.42 | 20.42 |

## 5.6 Visualization

In order to gain more insight, we present visualizations of the facebook107 dataset in Fig. 2(a). To demonstrate that our model can be applied to large networks, we present results of vGraph on a co-authorship network with around 100,000 nodes and 330,000 edges in Fig. 2(b). More visualizations are available in appendix B. We can observe that the community structure, or "social context", is reflected in the corresponding node embedding (node positions in both visualizations are determined by t-SNE of the node embeddings). To demonstrate the hierarchical extension of our model, we visualize a subset of the co-authorship dataset in Fig. 3. We visualize the first-tier communities and second-tier communities in panel (a) and (b) respectively. We can observe that the second-tier communities grouped under the same first-tier communities interact more with themselves than they do with other second-tier communities.

## 6 Conclusion

In this paper, we proposed vGraph, a method that performs overlapping (and non-overlapping) community detection and learns node and community embeddings at the same time. vGraph casts the generation of edges in a graph as an inference problem. To encourage collaborations between community detection and node representation learning, we assume that each node can be represented by a mixture of communities, and each community is defined as a multinomial distribution over nodes. We also design a smoothness regularizer in the latent space to encourage neighboring nodes to

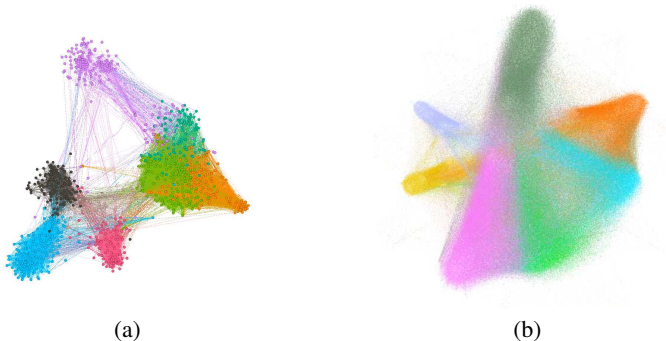

(a)                                                    (b)

Figure 2: In panel (a) we visualize the result on the facebook107 dataset using vGraph. In panel (b) we visualize the result on Dblp-full dataset using vGraph. The coordinates of the nodes are determined by t-SNE of the node embeddings.

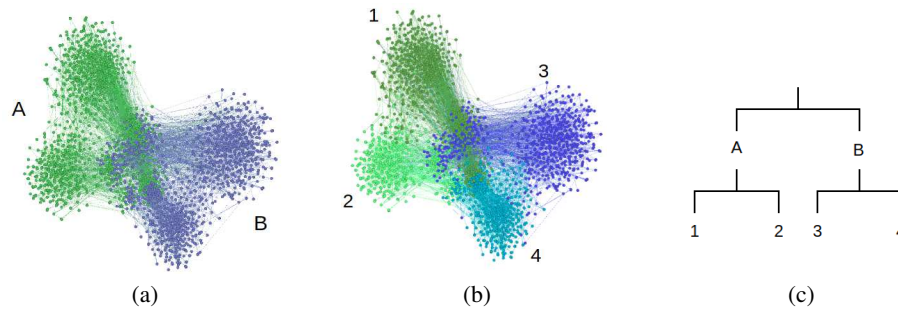

Figure 3: We visualize the result on a subset of Dblp dataset using two-level hierarchical vGraph. The coordinates of the nodes are determined by t-SNE of the node embeddings. In panel (a) we visualize the first-tier communities. In panel (b), we visualize the second-tier communities. In panel (c) we show the corresponding hierarchical tree structure.

be similar. Empirical evaluation on 20 different benchmark datasets demonstrates the effectiveness of the proposed method on both tasks compared to competitive baselines. Furthermore, our model is also readily extendable to detect hierarchical communities.

## Acknowledgments

This project is supported by the Natural Sciences and Engineering Research Council of Canada, as well as the Canada CIFAR AI Chair Program.

## Footnotes

[1]https://github.com/fanyun-sun/vGraph

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
