[Supplementary Material]

# vGraph: A Generative Model for Joint Community Detection and Node Representation Learning – Supplementary Material

## A Datasets

Citeseer, Cora, Cornell, Texas, Washington, and Wisconsin are available online[1]. For Youtube, Amazon, and Dblp, we consider subgraphs with the 5 largest ground-truth communities due to the runtime of the baseline methods.

**Facebook**[2] is a set of Facebook ego-networks. It contains 10 different ego-networks with identified circles. Social circles formed by friends are regarded as ground-truth communities.

**Youtube**[3] is a network of social relationships of Youtube users. The vertices represent users; the edges indicate friendships among the users; the user-defined groups are considered as ground-truth communities.

**Amazon**[4] is collected by crawling amazon website. The vertices represent products and the edges indicate products frequently purchased together. The ground-truth communities are defined by the product categories on Amazon.

**Dblp**[5] is a co-authorship network from Dblp. The vertices represent researchers and the edges indicate co-author relationships. Authors who have published in a same journal or conference form a community.

**Coauthor-CS**[6] is a computer science co-authorship network. We chose 21 conferences and group them into five categories: *Machine Learning*, *Computer Linguistics*, *Programming language*, *Data mining*, and *Database*.

 # B  Visualization

Figure 1: Visualization of the result of vGraph on the facebook1684 dataset. The coordinates of the nodes are determined by t-SNE of the node embeddings.

Figure 2: Visualization of the result of vGraph on the facebook107 dataset. The coordinates of the nodes are determined by t-SNE of the node embeddings.

Figure 3: Visualization of the result of vGraph on the facebook414 dataset. The coordinates of the nodes are determined by t-SNE of the node embeddings.

Figure 4: Visualization of the result of vGraph on the Youtube dataset. The coordinates of the nodes are determined by t-SNE of the node embeddings.

## Footnotes

[1] https://linqs.soe.ucsc.edu

[2] https://snap.stanfod.edu/data/ego-Facebook.html

[3] http://snap.stanford.edu/data/com-Youtube.html

[4] http://snap.stanford.edu/data/com-Amazon.html

[5] http://snap.stanford.edu/data/com-DBLP.html

[6] https://aminer.org/aminernetwork