[Reviews · NeurIPS 2019]

Reviewer 1



I would like to thank the authors for their response. ============================= Originality ----------- * Although the literature in node representation learning and community detection is quite vast, this paper is positioned in a way that diverts from the norm. The method proposed is treating the two tasks as one, and jointly learns embeddings, rather than going through iterations of optimizations. * The work includes an adequate number of citations to recent and older works in the tasks at hand. Moreover, the authors proceed to compare against a big number of these methods in the experimental section. * The novelty of the work is still limited, since the techniques used are well-known and have been applied to other domains already. Still, this work is the first to apply and combine them for community detection and node representation learning. Quality -------- * The theoretical analysis in this paper is quite shallow, since the techniques that are being used are already known and established. * The experimental section of the paper is one of the strongest points of this work. It is very extensive, compares the proposed work against a variety of other methods, and the comparison is done on a number of different tasks and using different metrics. * The authors are honest in evaluating their work, especially in Table 3. They still provide a reasonable explanation for the performance of their method compared to either ComE, or the rest of the methods. Clarity ------- * The paper is overall well-written; it feels like a polished work. * There are places that could be improved in terms of details, which I will mention below. Significance -------------- * Looking at novelty, this paper's strong points are the methodological contribution that it makes through the proposed modeling, and the kind of empirical analyses that it enables. In the rest of the dimensions, i.e., theoretical and algorithmic, the paper is not adding anything novel to the literature. * The proposed model enables a unique approach to data exploration. Indeed, as showcased in the plots, the learnt embeddings can be used to produce very interesting plots for quite large and messy networks. * As I mentioned previously, this method is a significant and highly non-trivial addition to the tools for network exploration. These tools are finding more and more users and use cases, which means that the potential impact of this work is high.

Reviewer 2



The joint analysis of node representations and community structure is an interesting and original avenue for analyzing networks. I agree with the authors that this should be much further investigated in the future, and indeed that more should be done in the networks realm in the area of integrative analysis. The comparison with current community detection and node representation algorithms was also very useful, and making the code publicly available is beneficial. The paper would have benefited from considering theoretical guarantees that are available to community detection methodologies, such as the detectability limits of the method as well as asymptotic properties of the method. Also, it wasn't entirely clear from the applications what the additional benefit was to applying vGraph over applying community detection and node embedding methods individually.

Reviewer 3



- The paper proposes a unifying framework, named vGraph (note: there are a few other, unrelated, approaches in the literature that share this moniker), for simultaneously pursuing node representation learning and community detection. The key insight is two learn two embeddings per node, one which captures the community assignment, and one which captures the graph topology. Moreover, a simultaneously learned embedding of the communities ties these two embeddings together in the probabilistic framework. - You are implicitly defining communities based on assortativity. It would be interesting to explore whether different choices of $\alpha_{w,c}$ and $d$ in (9) would allow for other community structures (e.g., disassortative communities or core-periphery communities) to be uncovered. -In all of your examples, $K$ (the number of communities) is relatively small and the communities recovered (at least upon first inspection) appear to be relatively balanced. Is your method effective in situations with many small communities (large $K$)? - For your methodology, when an oracle $K$ is unknown, how is it chosen? - In Section 2, subsection "Community Detection," you cover two groups of methods, matrix factorization-based methods and model-based methods, and claim that the computational complexity of these approaches is high. There are a bevy of other approaches (e.g., modularity optimization methods such as the Louvain method and Clauset, Newman and Moore's greedy modularity maximizing algorithm) that scale well and show excellent performance on real data applications. Moreover, the Louvain method (from Blondel et al. 2008) runs on graphs with order $>10^8$, and thin SVD based matrix factorizations can be applied to networks with millions of nodes using only relatively standard grid-based computing resources. - In deriving your probabilistic model and ELBO lower bound, you only consider a single $(c,w)$ pair. How are you aggregating over all edges? The model effectively extracts two node embeddings, one capturing communities and one capturing the edge structure, with the community embeddings tying the two together. Are the final node embedding $\phi$, $\varphi$ or a concatenation of the two? - Please list citations for the real data networks used in your experiments in the main text. The networks cited from [1] in the supplementary do not appear to be available at [1] anymore. -On the real data networks, what is the runtime of vGraph (for example, on Dblp-full) and competing models? There is no mention of the scalability of your approach. -Can you modify vGraph to incorporate nodal attributes? - Regarding Table 3, both NMI and Modularity can be computed for clusterings with $K'\neq K$. In the setting when $K$ is unknown, how would your approach compare to approaches that automate this model selection (i.e., the Louvain method)? - Often, clustering is an inference task with multiple (equally correct) truths (see, for example, Peel, L., Larremore, D. B., & Clauset, A. (2017). The ground truth about metadata and community detection in networks. Science advances, 3(5)). It is hard to judge the effectiveness of your approach based on its ability to recover a stated "ground truth," when the communities recovered by competing algorithms may be equally correct for recovering different (still true) communities. - In the classification experiment, what classifier are you using on the node-embeddings? (minor typo: vGraph is not best performing on 9 out of 12 datasets; there are only 6 datasets).

[Author Response · NeurIPS 2019]

We would like to thank each of the reviewers for reading our manuscript and providing very useful feedback. For minor
comments such as typos, missing citations, and choices of words/notation, we have fixed them. The reviewers raised a
few points that warrant more discussion than we are able to fit in the manuscript with the space constraints. Therefore,
we have expanded the supplementary section in the revised version. Below, we address the key points in detail.

**R3: Runtime and scalability.** Here we provide the runtime of vGraph and the two fastest overlapping community
detection baselines on DBLP-full: vGraph trained for 5000 iterations with batch size 1000 (277s), BigCLAM (193s),
CESNA (1191s). Note that it is not a fair comparison as our code is in Python whereas the other codes are implemented
in C++. Below we further show the time complexity of vGraph. Sampling an edge takes constant time, thus calculating
Eq. 4 takes $O(d(M + 1))$ time, where $M$ is the number of negative samples and $d$ is the dimension of embeddings (the
node embeddings and community embeddings have the same dimension). To calculate Eq. 6, it takes $O(dK)$ time
where $K$ is the number of communities. Thus, an iteration with one sample takes $O(\max(dM, dK))$ time. In practice
the number of updates required is proportional to the number of edges $O(|\mathcal{E}|)$, thus the overall time complexity of
vGraph is $O(|\mathcal{E}|d\max(M, K))$. The fact that (1) it scales linearly to the number of edges and (2) it employs negative
sampling combined with batch-wise stochastic optimization makes vGraph scalable. We will add this analysis in the
revised version. **R3: Regarding the aggregation over edges** We aggregate over all edges by taking summation of the
ELBO bound on each edge.

**R2: Benefits of vGraph.** From Tables 2, 3, and 4 we can see that vGraph outperforms methods that perform community
detection or node representation individually since vGraph integrates the two tasks, which are beneficial to each other.
Moreover, vGraph is efficient and scalable compared to classical community detection methods.

**R1&R3: Regarding the design of the two sets of node embeddings.** R1: Recall our generative process: (1) we first
draw a community assignment $z \sim p(z|w)$ representing the social context of node $w$, (2) then based on $z$, we generate
a neighbor of $w$, ($c$ can be referred to as the "context" of $w$) $c \sim p(c|z)$. The first set of node embeddings is used in
step (1) and the second set of node embeddings is used for step (2) in the generation process. A similar concept is used
in existing node representation learning methods (e.g. DeepWalk, LINE, node2vec) and matrix factorization methods
where they use two different sets of node embeddings. R3: Note that in vGraph, the two set of embeddings are tied
together by community embeddings and thus capture similar information. For the final embedding, we take the first
set of node embeddings ($\phi$). In fact, we can also share parameters for two embeddings (that is, $\phi = \varphi$) and it yielded
similar performance. We have updated the manuscript to make this more clear.

**R1&R3: Design of smoothness regularization**. R1: For the distribution distance $d$, we did experiment with divergence
based distance metrics and found they did not yield much difference. Thus, we used squared difference as in [25] for
the sake of simplicity. R3: Indeed, by designing smoothness regularization the way it is in the paper, we are implicitly
considering communities based on assortativity. We agree that it is worth exploring different forms of smoothing
functions that possibly favor detecting different kinds of communities. For now, we left this as future work.

**R3: Experiment settings and design.** We agree that there are many efficient approaches to community detection, as
you pointed out in the review, and covering all of them is difficult. **(1) About baselines.** In fact, our experiments are
designed to demonstrate that the vGraph framework enables community detection and node representation learning
to benefit one other, not to prove that it outperforms all existing studies. Therefore, we decided to choose certain
representative methods (i.e., matrix factorization-based methods, generative models, and K-Means after node embed-
dings) which help validate this point. We will discuss more studies in the revised draft. **(2) About choosing $K$.** In
practice, when the true $K$ is not given, we can still choose $K$ according to the performance on validation set (as in
[14,36]). However, existing studies typically assume that the oracle $K$ is given in experiments [4,18,25,30,32]. We
follow the same experiment setting. Also, those parametric models compare only with other parametric methods, not
non-parametric models such as Louvain. In practice, we can still compare with non-parametric methods but we have to
make sure comparisons are made under the same model complexity (i.e. the same number of communities). **(3) About
cases with many small communities.** Since communities are determined based on node embeddings, the algorithm
works regardless of the number of communities. **(4) On evaluation on ground truth communities.** We agree that
there may be multiple equally correct ground truths in practice. However, all the datasets we consider only have one
"ground-truth" label. Thus, testing community detection algorithms on the given ground-truth is the best we can do
and it is still the most widely adopted way to evaluate community detection [4,14,18,20,36,37]. Furthermore, we use
modularity as one of the evaluation metrics in our experiments (Table 3) and modularity does not depend on ground
truth communities. **(5) In the classification experiment, what classifier are you using on the node-embeddings?**
We employ an one-vs-rest logistic regression classifier using the commonly used LIBLINEAR package.

**R3: Can you modify vGraph to incorporate nodal attributes?** vGraph is more of a principled framework to
integrate community detection and node representation learning, where it is flexible to use different kinds of encoders
for learning node representations. To incorporate node attributes, we can simply use graph neural networks (e.g., Graph
Convolutional Networks) as the node encoder.

[Meta-Review · NeurIPS 2019]

The idea of combining community detection and learning node representation is quite natural but apparently it is not well entrenched in the network analysis community. The reviewers feel that for this reason the paper would be a valuable contribution to the field.